DATA RELEASE

# Genomic identification, annotation, and comparative analysis of Vacuolar-type ATP synthase subunits in *Diaphorina citri*

Rebecca Grace[1,2], Crissy Massimino[1], Teresa D. Shippy[3], Will Tank[3], Prashant S. Hosmani[4], Mirella Flores-Gonzalez[4], Lukas A. Mueller[4], Wayne B. Hunter[5], Joshua B. Benoit[6], Susan J. Brown[3], Tom D'Elia[1] and Surya Saha[4,7,*]

1   Indian River State College, Fort Pierce, FL 34981, USA
2   Department of Molecular Biology and Genetics, Cornell University, Ithaca, NY 14853, USA
3   Division of Biology, Kansas State University, Manhattan, KS 66506, USA
4   Boyce Thompson Institute, Ithaca, NY 14853, USA
5   USDA-ARS, US Horticultural Research Laboratory, Fort Pierce, FL 34945, USA
6   Department of Biological Sciences, University of Cincinnati, Cincinnati, OH 45221, USA
7   Animal and Comparative Biomedical Sciences, University of Arizona, Tucson, AZ 85721, USA

## ABSTRACT

The hemipteran insect *Diaphorina citri*, or Asian citrus psyllid, is a vector for *Candidatus* Liberibacter asiaticus (*C*Las), the bacterium causing citrus greening disease, or Huanglongbing (HLB). Millions of citrus trees have been destroyed, and every grove in Florida, USA, has been directly affected by this disease. In eukaryotes, vacuolar-type ATP synthase (V-ATPase) is an abundant heterodimeric enzyme that serves the cell with essential compartment acidification through the active processes that transport protons across the membrane. Fifteen putative *V-ATPase* genes in the *D. citri* genome were manually curated. Comparative genomic analysis revealed that *D. citri* V-ATPase subunits share domains and motifs with other insects, including the V-ATPase-A superfamily domain. Phylogenetic analysis separates *D. citri* V-ATPase subunits into expected clades with orthologous sequences. Annotation of the *D. citri* genome is a critical step towards developing directed pest management strategies to reduce the spread of HLB throughout the citrus industry.

**Submitted:**   21 October 2021

\* Corresponding author. E-mail: suryasaha@cornell.edu

Preprint submitted at https://doi.org/10.1101/2021.10.18.464890

Included in the series: ***Asian citrus psyllid community annotation*** (https://doi.org/10.46471/GIGABYTE_SERIES_0001)

**Subjects**   Genetics and Genomics, Animal Genetics, Bioinformatics

## INTRODUCTION

Vacuolar (H⁺)-ATP synthase (V-ATPase) is a highly conserved eukaryotic enzyme [1]. Originally identified in the vacuole membrane, V-ATPase has a critical function in the plasma membrane and endomembrane system of almost every cell [2, 3]. V-ATPase regulates the acidity of organelles, such as vacuoles, the Golgi apparatus, and coated vesicles, by translocating protons across their membranes and powering secondary transport processes. Structurally, V-ATPase has a noncatalytic transmembrane domain, the $V_0$ rotor, and a catalytic cytoplasmic domain, the $V_1$ stator. V-ATPase hydrolyzes adenosine triphosphate (ATP) into adenosine diphosphate (ADP), thus acting in the opposite manner to

the related F-ATPase [1]. In insects, 13 protein subunits are typically required to build a single V-ATPase [4]. The $V_0$ domain comprises subunits a–e, and $V_1$ comprises subunits A–H [5]. The critical accessory subunit S1 (Ac45) also helps assemble the enzyme [6].

## CONTEXT

In insects, high levels of V-ATPase are found in epithelial cells. They are especially important in the digestive tract, helping to regulate nutrient uptake and solute transport [7]. Studies in several phyla, including insects, have demonstrated the lethality of silencing individual *V-ATPase* genes, making *V-ATPase* an attractive target for RNA interference (RNAi)-based pest control [1, 4, 7]. We have characterized the genes encoding V-ATPase subunits in *Diaphorina citri* (Hemiptera: Liviidae; NCBI:txid121845) as a step towards the development of future management strategies to reduce the psyllid vector of *Candidatus* Liberibacter asiaticus (*C*Las), the causative bacterial agent of Huanglongbing (HLB), also known as citrus greening disease.

## METHODS

V-ATP synthase insect orthologs from *Acyrthosiphon pisum* (pea aphid) were obtained from the Kyoto Encyclopedia of Genes and Genomes (KEGG) database (RRID:SCR_012773). Additional ortholog subunits occurring in non-insect eukaryotes, like human (*Homo sapiens*), were obtained from the HUGO Gene Nomenclature Committee (HGNC) (RRID:SCR_002827) and the nonredundant National Center for Biotechnology Information (NCBI) Reference Sequence (Ref-Seq) database [8]. V-ATPase protein sequences were used to query the predicted protein set from the *D. citri* MCOT (Maker (RRID:SCR_005309), Cufflinks (RRID:SCR_014597), Oases (RRID:SCR_011896), and Trinity (RRID:SCR_013048)) transcriptome via protein BLAST (BLASTp) [9]. Reciprocal BLASTp analysis was performed to validate the *D. citri* MCOT significant hits using the NCBI nonredundant protein database [8]. *D. citri V-ATPase* genes were identified in the genome (version 1.91) by searching for the identified mapped MCOT models in the WebApollo (RRID:SCR_005321) system hosted at the Boyce Thompson Institute. Multiple alignments of the predicted *D. citri* MCOT proteins, other gene model sequences, and insect *V-ATPase* orthologs were performed using the European Bioinformatics Institute MUSCLE alignment online tool (RRID:SCR_004727) [10]. Further analysis using RNA sequencing (RNA-seq) reads, Illumina DNA sequencing reads, StringTie models, and Pacific Biosciences Isoform sequencing transcripts were used to manually annotate the final *V-ATPase* gene models. Manually annotated *V-ATPase* gene models were then integrated into the version 3.0 official gene set (OGS). *V-ATPase* genes were verified in WebApollo through analysis using *de novo*-assembled transcripts, Iso-seq transcripts, Augustus models, Mikado transcriptome, SwissProt proteins, and SNAP prediction models. A list of annotated *D. citri* identifiers and a sample of evidence supporting the annotated models is found in Table 1. A more detailed description of the annotation workflow is available (Figure 1) [11]. V-ATPase nomenclature is somewhat inconsistent in the literature and between species, therefore, we have used nomenclature standards reported in previous work on other Hemiptera [12, 13].

Reciprocal BLASTp of manually annotated v3.0 *V-ATPase* genes were performed at the NCBI comparing the Insecta taxid. Insect orthologs from *Acyrthosiphon pisum* (pea aphid) [14, 15], *Bemisia tabaci* (whitefly) [16], *Aedes aegypti* (yellow fever mosquito) [17], *Apis mellifera* (honeybee) [18], *Tribolium castaneum* (red flour beetle) [19], and *Drosophila*

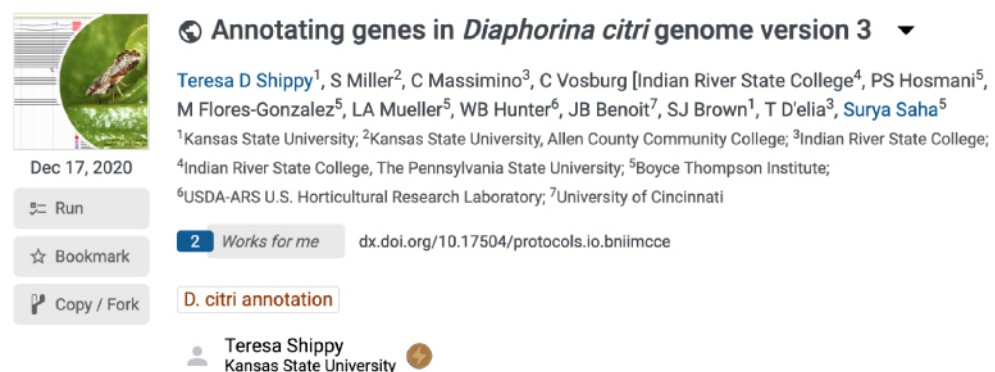

**Figure 1.** Protocols.io protocol for psyllid genome curation [11]. https://www.protocols.io/widgets/doi?uri=dx.doi.org/10.17504/protocols.io.bniimcce

**Table 1.** Evidence for gene annotations. MCOT transcriptome identifiers included, if applicable.

| Gene | Identifier | MCOT | *de novo* transcriptome | Iso-seq | RNA-seq | Ortholog |
|---|---|---|---|---|---|---|
| *V-ATPase a1* | Dcitr07g04330.1.1 | MCOT05340.0.CO | X | X | X | X |
| *V-ATPase a2* | Dcitr07g01670.1.1 | MCOT20572.0.CT | X | X | X | X |
| *V-ATPase b* | Dcitr12g08560.1.1 | - | X | X | X | X |
| *V-ATPase c* | Dcitr06g11110.1.1 | - | X | X | X | X |
| *V-ATPase d* | Dcitr04g06930.1.1 | - | - | X | X | X |
| *V-ATPase e* | Dcitr03g19730.1.1 | - | X | X | X | X |
| *V-ATPase A* | Dcitr06g09110.1.1 | MCOT04747.0.CC | - | X | X | X |
| *V-ATPase B* | Dcitr09g08730.1.1 | - | X | X | X | X |
| *V-ATPase C* | Dcitr02g01535.1.1 | - | X | X | X | X |
| *V-ATPase D* | Dcitr09g02030.1.1 | - | - | X | X | X |
| *V-ATPase E* | Dcitr04g09575.1.1 | - | - | X | X | X |
| *V-ATPase F* | Dcitr07g06920.1.1 | MCOT14638.0.CC | X | X | X | X |
| *V-ATPase G* | Dcitr11g08810.1.1 | MCOT22289.0.CT | - | X | X | X |
| *V-ATPase H* (partial, N-terminus) | Dcitr00g06320.1.1 | - | - | - | - | - |
| *V-ATPase H* (partial, C-terminus) | Dcitr01g01240.1.1 | MCOT00604.0.CT | - | X | X | X |
| *V-ATPase Ac45* | Dcitr09g09620.1.1 | MCOT16252.0.CC | X | X | X | X |

MCOT: Maker, Cufflinks, Oases, Trinity.

*melanogaster* (fruit fly) [20] were obtained by reciprocal BLASTp (RRID:SCR_004870) analysis of the nonredundant protein database at NCBI [8]. A neighbor-joining phylogenetic tree was constructed using MUSCLE (RRID:SCR_011812) multiple sequence alignment with the Poisson correction method and 1000 replicate bootstrap test using full-length protein sequences in MEGA version 7 (RRID:SCR_000667) for the transmembrane complex, the catalytic complex, and the accessory subunit Ac45, respectively (Figures 2–4) [21]. The sequence accession numbers used in these analyses can be found in Tables 2–4. Comparative expression levels of *D. citri V-ATPases* throughout egg, nymph, and adult life stages in *D. citri* insects both exposed and not exposed to *C*Las were determined using RNA-seq data and the Citrus Greening Expression Network (CGEN) [9]. These gene expression levels were visualized using the pheatmap package in R (RRID:SCR_016418) [22, 23]. Expression values for all samples discussed in this manuscript are visualized in Figures 5 and 6 and are reported as transcripts per million (TPM) in Table 5.

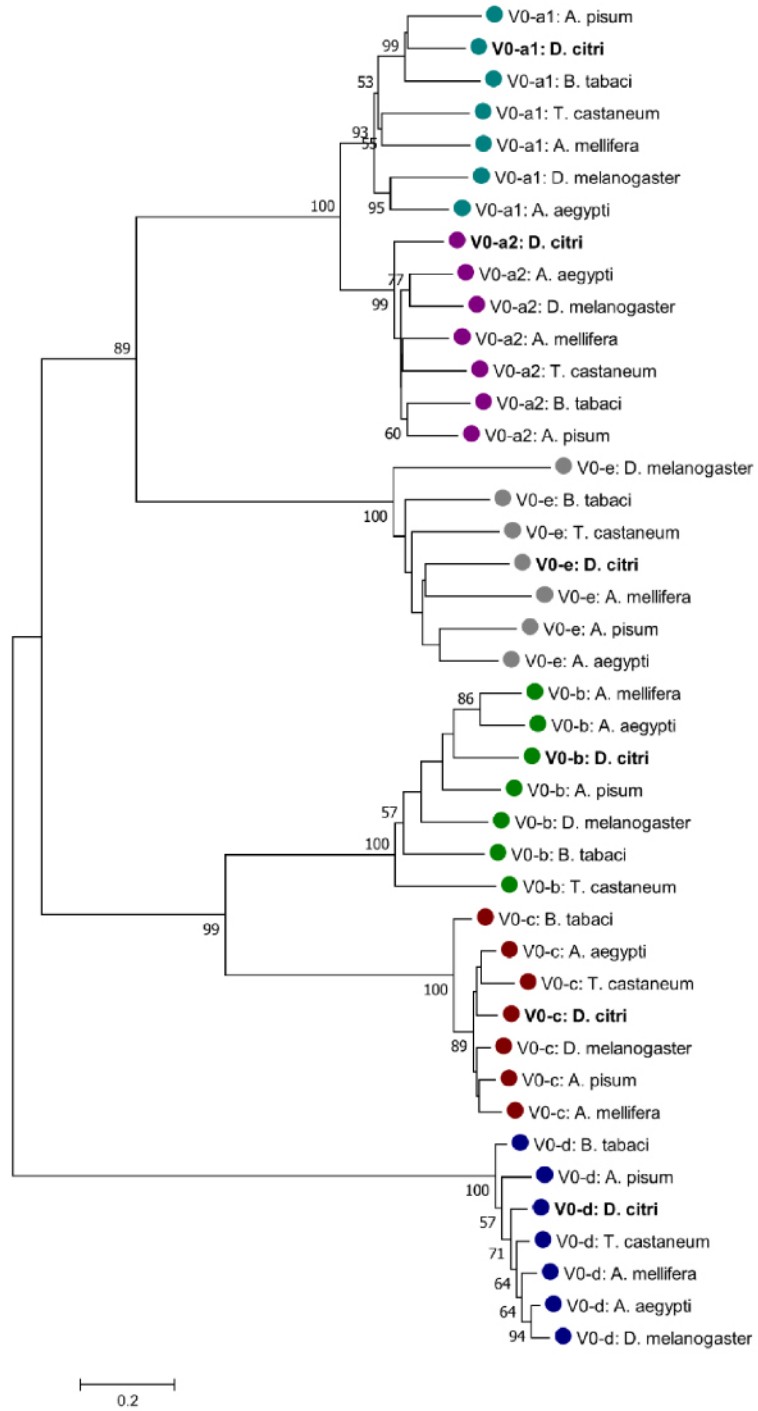

**Figure 2.** Phylogenetic analysis of V-ATPase $V_0$, transmembrane domain, subunits a–e. The tree was constructed with MEGA7 software [21] using MUSCLE for alignment of amino acid sequences, followed by neighbor-joining analysis with 1000 bootstrap replications. Values greater than 50 are shown at nodes. *Diaphorina citri* is marked in bold and color-coding indicates specific $V_0$ subunit groups. NCBI accession numbers are shown in Table 2.

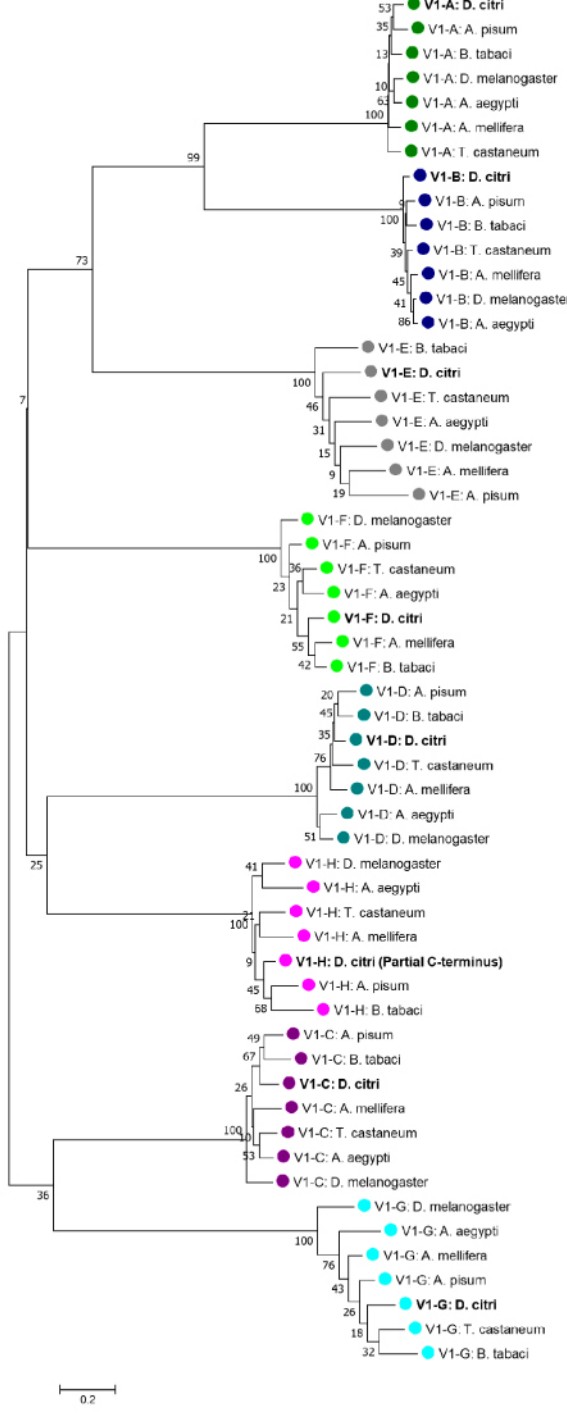

**Figure 3.** Phylogenetic analysis of V-ATPase $V_1$, catalytic domain, subunits A-G. The tree was constructed with MEGA7 [21] software using MUSCLE for alignment of amino acid sequences, followed by neighbor-joining analysis with 1000 bootstrap replications. Values greater than 50 are shown at nodes. *Diaphorina citri* is marked in bold and color-coding indicates specific $V_1$ subunit groups. *D. citri* V-ATPase H ($V_1$-H) was annotated as two partial gene models, therefore, only the C-terminus, partial amino acid sequence of $V_1$-H was included in this analysis. NCBI accession numbers are shown in Table 3.

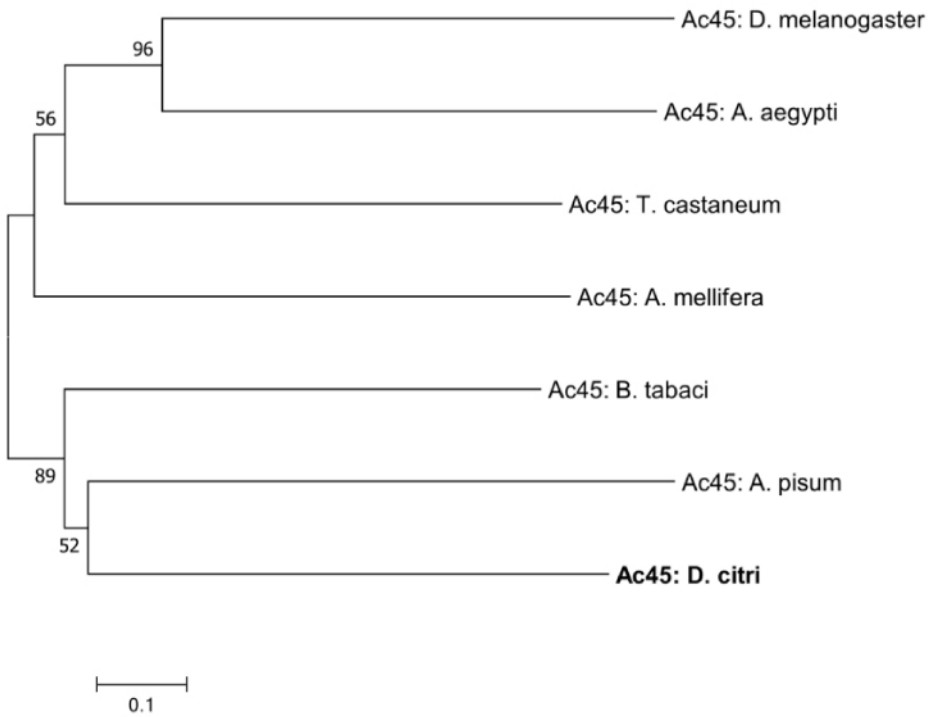

**Figure 4.** Phylogenetic analysis of V-ATPase accessory subunit Ac45. The tree was constructed with MEGA7 software [21] using MUSCLE for alignment of amino acid sequences, followed by neighbor-joining analysis with 1000 bootstrap replications. Values greater than 50 are shown at nodes. *Diaphorina citri* is marked in bold. NCBI accession numbers are shown in Table 4.

## DATA VALIDATION AND QUALITY CONTROL

Genes encoding all 13 subunits required to build a single V-ATP synthase enzyme, as well as an accessory subunit S1 (*Ac45*) gene, were annotated in *D. citri*. There were no additional subunits found in *D. citri*, as reported in other metazoans [2]. Although insect V-ATPases are known to contain 13 subunits, there is variation in the gene copy number for individual subunits among different species (Tables 6–8). The $V_0$ transmembrane domain subunits *V-ATPase a, b*, and *e*; the $V_1$ catalytic domain subunits *V-ATPase A, C, D,* and *G*; and *Ac45*, all show variation in copy number among different species. The three Hemipterans analyzed (*D. citri*, *A. pisum*, and *B. tabaci*) maintain the same paralog number for all *V-ATPase* genes, except for *A. pisum V-ATPase D* and *G*, compared with the other orders (Table 7). This variation in copy number is interesting in contrast to the genes *V-ATPase c, d, B, E, F*, and *H*, which maintain only one gene copy across all orders of Insecta found in Tables 6–8. V-ATPase subunits have been studied in plants, animals, fungi, and insects, and certain genes have been highlighted for their functional versatility in serving cell needs. For example, yeast and mammals have numerous copies and isoforms of the transmembrane proteolipid *V-ATPase a* with functions supporting vacuoles, Golgi, neurons, osteoclasts, and epididymal cells [2]. *D. citri,* along with *A. pisum* and *B. tabaci*, has two copies of the *V-ATPase a* gene, whereas *D. melanogaster* has five copies (Table 6). In *D. citri*, a paralog of *V-ATPase a* was found, and they maintain differences in their amino acid sequences (Tables 2, 6) [24]. Phylogenetic analysis of $V_0$ subunit protein sequences supports that the

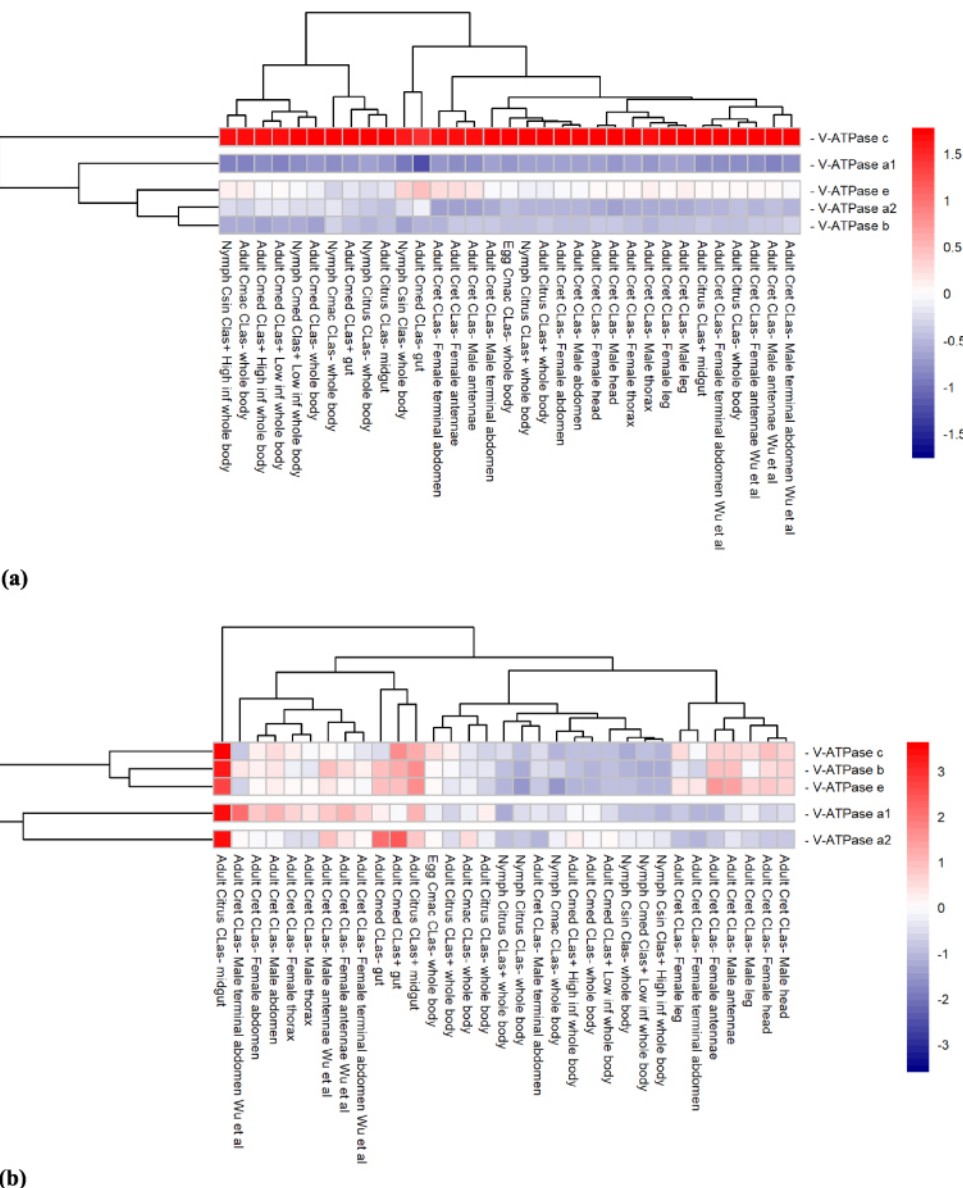

**Figure 5.** Comparative expression levels of the *Diaphorina citri* V-ATPase genes encoding the $V_0$, transmembrane, subunits in *D. citri* insects reared on various infected and uninfected citrus varieties. Expression data were collected from the Citrus Greening Expression Network [9], with psyllid RNA-seq data from NCBI BioProjects PRJNA609978 and PRJNA448935, in addition to several published datasets [25–29]. Citrus hosts are abbreviated as *Csin* (*Citrus sinensis*), *Cmed* (*Citrus medica*), *Cret* (*Citrus reticulata*), and *Cmac* (*Citrus macrophylla*). Transcripts per million (TPM) values are listed in Table 5. Rows of genes and columns of RNA-seq data are clustered based on expression differences. (a) Expression data scaled by sample. (b) Expression data scaled by gene.

duplication event occurred before the divergence of Hemimetabola and Holometabola (Figure 2).

We identified complete genes in genome v3.0 for all the subunits except *V-ATPase H*. Using genome-independent transcript sequences [24], we were able to determine that the 3′ portion of the *V-ATPase H* gene is located on chromosome 1, but the 5′ end of the gene is on

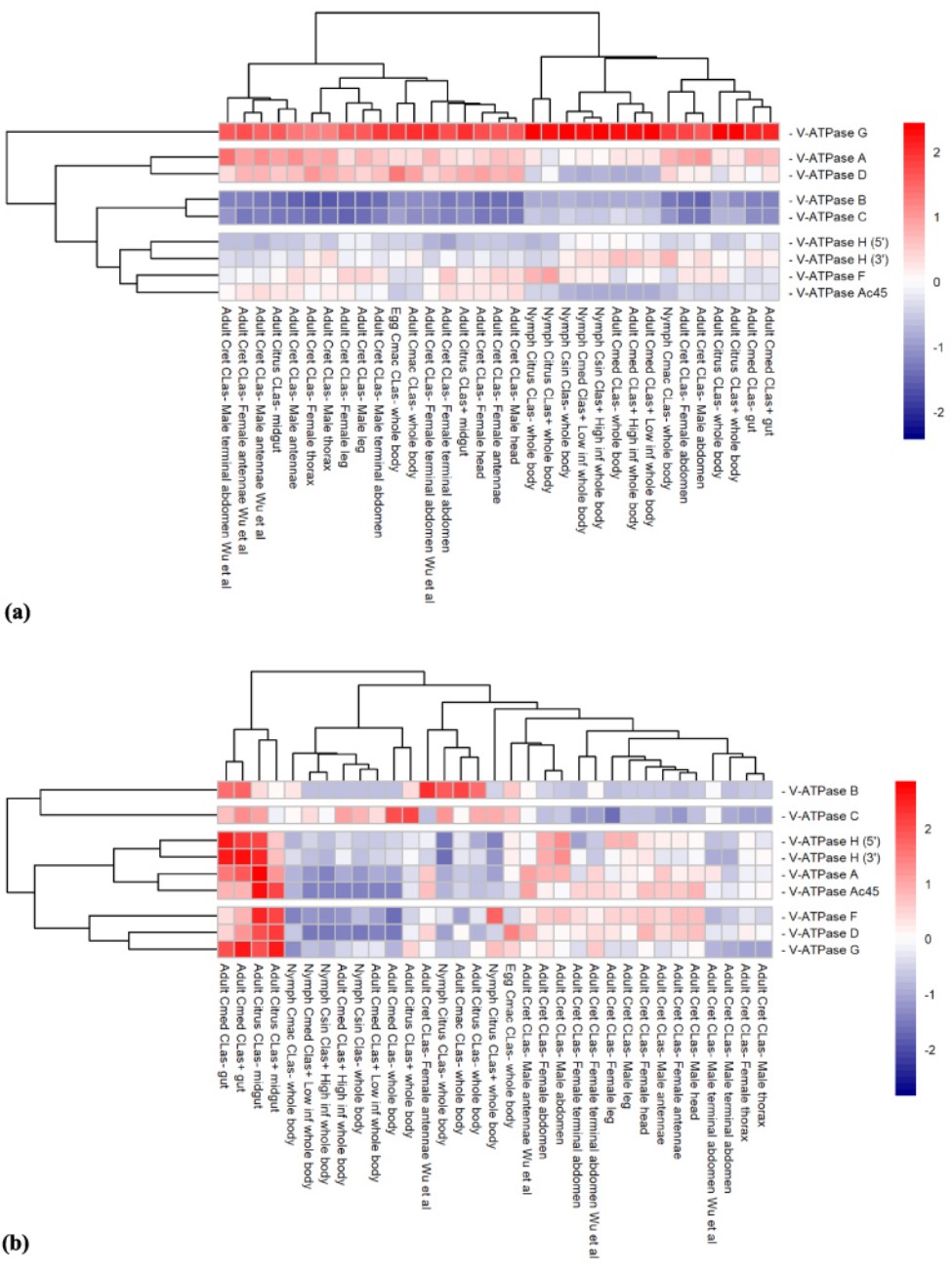

**Figure 6.** Comparative expression levels of the *Diaphorina citri* V-ATPase genes encoding the $V_1$, catalytic, and accessory subunits in *D. citri* insects reared on various infected and uninfected citrus varieties. V-ATPase H was annotated as two partial models and are both represented here separately as V-ATPase H (5′), denoting the 5-prime end of the gene, and V-ATPase H (3′), denoting the 3-prime end of the gene. Expression data were collected from the Citrus Greening Expression Network [9], with psyllid RNA-seq data from NCBI BioProjects PRJNA609978 and PRJNA448935, in addition to several published datasets [25–29]. Citrus hosts are abbreviated as *Csin* (*Citrus sinensis*), *Cmed* (*Citrus medica*), *Cret* (*Citrus reticulata*), and *Cmac* (*Citrus macrophylla*). Transcript per million (TPM) values are listed in Table 5. Rows of genes and columns of RNA-seq data are clustered based on expression differences. (a) Expression data scaled by sample. (b) Expression data scaled by gene.

**Table 2.** *V-ATPase* transmembrane subunit ($V_0$) BLAST table.

| Gene | | *A. pisum* | *B. tabaci* | *T. castaneum* | *D. melanogaster* | *A. aegypti* | *A. mellifera* |
|---|---|---|---|---|---|---|---|
| *V-ATPase a1* | Accession | XP_029343960.1 | XP_018903507.1 | XP_008200952.1 | NP_650722.1 | XP_021706364.1 | XP_006565533.1 |
| | Bit score | 1332 | 1312 | 1128 | 1151 | 1217 | 1166 |
| | QC (%) | 99 | 100 | 100 | 99 | 99 | 99 |
| | Identity (%) | 75.21 | 72.63 | 66.59 | 66.23 | 69.75 | 68.56 |
| *V-ATPase a2* | Accession | XP_008183003.1 | XP_018913655.1 | XP_008200806.1 | NP_733274.1 | XP_021693139.1 | XP_026298707.1 |
| | Bit score | 1369 | 1334 | 1331 | 1286 | 1369 | 1392 |
| | QC (%) | 100 | 100 | 100 | 100 | 100 | 100 |
| | Identity (%) | 79.10 | 76.42 | 76.18 | 76.19 | 77.21 | 79.05 |
| *V-ATPase b* | Accession | NP_001155679.1 | XP_018909463.1 | NP_001161226.1 | NP_001247111.1 | XP_001662256.1 | XP_392599.1 |
| | Bit score | 294 | 283 | 231 | 270 | 275 | 300 |
| | QC (%) | 99 | 99 | 98 | 99 | 99 | 99 |
| | Identity (%) | 73.79 | 73.30 | 63.05 | 72.95 | 74.4 | 76.33 |
| *V-ATPase c* | Accession | NP_001155531.1 | XP_018897791.1 | XP_967959.1 | NP_476801.1 | XP_001654757.1 | NP_001011570.1 |
| | Bit score | 268 | 250 | 258 | 271 | 267 | 259 |
| | QC (%) | 98 | 99 | 98 | 99 | 98 | 98 |
| | Identity (%) | 92.11 | 86.93 | 89.54 | 92.16 | 92.76 | 89.47 |
| *V-ATPase d* | Accession | NP_001191854.1 | XP_018903442.1 | XP_974905.1 | NP_570080.1 | XP_001661299.1 | XP_393438.2 |
| | Bit score | 656 | 669 | 671 | 661 | 677 | 679 |
| | QC (%) | 99 | 100 | 10 | 99 | 100 | 100 |
| | Identity (%) | 89.60 | 91.95 | 91.67% | 90.20 | 91.95 | 92.82 |
| *V-ATPase e* | Accession | XP_003242132.1 | XP_018909271.1 | XP_971898.1 | NP_001097499.1 | ABF18129.1 | XP_624787.1 |
| | Bit score | 127 | 119 | 123 | 106 | 133 | 127 |
| | QC (%) | 96 | 96 | 94 | 96 | 100 | 94 |
| | Identity (%) | 69.51 | 70.73 | 72.50 | 58.54 | 70.59 | 71.25 |

Accession number, bit score, query coverage (QC), and identity results from protein BLAST analysis of annotated *D. citri V-ATPase* transmembrane subunit genes to their putative orthologs.

one of the unplaced chromosomes that make up chromosome 0 (Table 1). Tables 2–4 show the results of protein BLAST analysis comparing the same insects as found in Tables 6–8. Other than *Ac45*, all subunits share a relatively high sequence identity, approximately 57–94%, among individual pairwise alignments with each *D. citri* sequence (Tables 2–3). BLAST results of annotated gene models had high query coverage to orthologs, supporting the completeness of the annotated gene models. In contrast, the sequence identities of *Ac45*, approximately 24–33%, show the highest divergence when comparing *D. citri* to other insects (Table 4). For the $V_0$, transmembrane domain, subunits in Table 2, proteolipid subunit c (*V-ATPase c*) maintains some of the highest percentages of sequence identity, highlighting the importance of the protein function to form the c-ring that rotates and ultimately translocates protons across various membranes [2]. This is supported by the data shown in Table 6, in which a single gene copy for *V-ATPase c* is maintained across different orders of insects.

The Citrus Greening Expression Network (CGEN) was used to compare transcript expression levels in various regions of *D. citri* that have either been exposed to or not exposed to *C*Las infection [9, 30, 31]. Figure 5 shows a heatmap comparing $V_0$ subunit expression levels found under various conditions. *V-ATPase c* is visually differentiated from other *V-ATPase* transmembrane subunit genes by its inflated expression levels (Figure 5a). *V-ATPase c* expression also shows a 2.63-fold increase, from 647.9 to 1705.25 TPM, in the guts of adult psyllids fed on infected *versus* uninfected *Citrus medica* leaves (Figure 5b, Table 5). These expression levels, coupled with the fundamental cellular nature and relatively even occurrence of *V-ATPases*, suggest that *V-ATPase* genes are good candidates for RNAi.

**Table 3.** *V-ATPase* catalytic subunit (V$_1$) BLAST table.

| Gene | | *A. pisum* | *B. tabaci* | *T. castaneum* | *D. melanogaster* | *A. aegypti* | *A. mellifera* |
|---|---|---|---|---|---|---|---|
| *V-ATPase A* | Accession | XP_008179407.1 | XP_018897790.1 | XP_976188.1 | NP_001246015.1 | XP_021709029.1 | XP_016769524.1 |
| | Bit score | 1176 | 1187 | 1167 | 1162 | 1168 | 1174 |
| | QC (%) | 99 | 99 | 99 | 99 | 100 | 99 |
| | Identity (%) | 91.84 | 92.47 | 90.21 | 90.18 | 90.57 | 90.72 |
| *V-ATPase B* | Accession | XP_003246082.1 | XP_018896879.1 | XP_967844.1 | NP_001163597.1 | XP_001651458.1 | XP_624112.1 |
| | Bit score | 966 | 976 | 966 | 966 | 968 | 955 |
| | QC (%) | 98 | 99 | 98 | 98 | 98 | 98 |
| | Identity (%) | 93.88 | 94.33 | 93.48 | 93.88 | 94.09 | 92.87 |
| *V-ATPase C* | Accession | XP_001946227.1 | XP_018915661.1 | XP_008195426.1 | NP_477266.1 | XP_021695404.1 | XP_006562159.1 |
| | Bit score | 702 | 653 | 671 | 645 | 671 | 648 |
| | QC (%) | 99 | 99 | 98 | 98 | 99 | 9 |
| | Identity (%) | 87.05 | 82.81 | 82.20 | 79.11 | 81.77 | 79.95% |
| *V-ATPase D* | Accession | NP_001119691.1 | XP_018904914.1 | XP_975872.1 | NP_651987.1 | XP_001660426.1 | XP_394769.2 |
| | Bit score | 420 | 423 | 426 | 408 | 401 | 404 |
| | QC (%) | 100 | 100 | 100 | 100 | 100 | 100 |
| | Identity (%) | 87.24 | 86.42 | 86.53 | 82.52 | 80.89 | 86.94 |
| *V-ATPase E* | Accession | NP_001155650.1 | XP_018901912.1 | XP_970621.1 | NP_001287182.1 | XP_001655825.1 | XP_625098.1 |
| | Bit score | 301 | 336 | 341 | 325 | 335 | 336 |
| | QC (%) | 99 | 100 | 100 | 100 | 100 | 100 |
| | Identity (%) | 73.21 | 71.68 | 73.01 | 71.24 | 70.80 | 73.01 |
| *V-ATPase F* | Accession | NP_001119690.1 | XP_018905603.1 | XP_975016.1 | NP_476969.1 | XP_001655376.1 | XP_624852.1 |
| | Bit score | 223 | 227 | 216 | 217 | 216 | 230 |
| | QC (%) | 98 | 98 | 98 | 98 | 100 | 98 |
| | Identity (%) | 85.25 | 88.52 | 82.79 | 81.15 | 82.26 | 88.52 |
| *V-ATPase G* | Accession | NP_001119628.1 | XP_018908074.1 | XP_973974.1 | NP_001287407.1 | XP_001652605.1 | XP_624346.1 |
| | Bit score | 184 | 188 | 173 | 154 | 171 | 180 |
| | QC (%) | 99 | 100 | 98 | 96 | 96 | 98 |
| | Identity (%) | 80.34 | 81.51 | 74.14 | 65.79 | 73.68 | 76.72 |
| *V-ATPase H* (partial, N-terminus) | Accession | XP_001949116.3 | XP_018904009.1 | NP_001280516.1 | NP_001260510.1 | XP_001652018.1 | XP_003251675.1 |
| | Bit score | 132 | 162 | 132 | 110 | 130 | 142 |
| | QC (%) | 90 | 99 | 99 | 84 | 91 | 94 |
| | Identity (%) | 57.69 | 68.42 | 57.26 | 56.70 | 58.10 | 64.22 |
| *V-ATPase H* (partial, C-terminus) | Accession | XP_001949116.3 | XP_018904009.1 | NP_001280516.1 | NP_001260510.1 | XP_001652018.1 | XP_003251675.1 |
| | Bit score | 530 | 506 | 513 | 493 | 498 | 490 |
| | QC (%) | 100 | 99 | 100 | 100 | 100 | 99 |
| | Identity (%) | 83.95 | 82.33 | 83.28 | 77.59 | 78.93 | 82.15 |

Accession number, bit score, query coverage (QC), and identity results from protein BLAST analysis of annotated *Diaphorina citri V-ATPase* catalytic subunit genes to their putative orthologs.

**Table 4.** *V-ATPase Ac45*, accessory subunit, BLAST table.

| Gene | | *A. pisum* | *B. tabaci* | *T. castaneum* | *D. melanogaster* | *A. aegypti* | *A. mellifera* |
|---|---|---|---|---|---|---|---|
| *Ac45* | Accession | NP_001162140.1 | XP_018899028.1 | XP_974187.2 | NP_610470.1 | XP_001658652.1 | XP_001121483.3 |
| | Bit score | 136 | 177 | 114 | 77.4 | 92.4 | 88.6 |
| | QC (%) | 83 | 97 | 100 | 96 | 95 | 93 |
| | Identity (%) | 30.87 | 33.49 | 27.03 | 24.94 | 23.81 | 26.35 |

Accession number, bit score, query coverage (QC), and identity results from protein BLAST analysis of annotated *Diaphorina citri V-ATPase Ac45* gene to its putative orthologs.

Silencing a V$_0$ transcript should have inhibitory effects on the assembly of the V-ATPase enzyme. In particular, if infected psyllids increase in their demand for higher *V-ATPase c* expression levels overall, knocking this transcript down is likely to be detrimental for the

**Table 5.** Expression values listed as transcripts per million (TPM), visualized in Figures 5 and 6.

| Gene name | V-ATPase a1 | V-ATPase a2 | V-ATPase b | V-ATPase c | V-ATPase e | V-ATPase A | V-ATPase B | V-ATPase C | V-ATPase D | V-ATPase F | V-ATPase G | V-ATPase H (5′) | V-ATPase H (3′) | V-ATPase Ac45 |
|---|---|---|---|---|---|---|---|---|---|---|---|---|---|---|
| Gene ID | Dcitr07 g04330.1.1 | Dcitr07 g01670.1.1 | Dcitr12 g08560.1.1 | Dcitr06 g11110.1.1 | Dcitr03 g19730.1.1 | Dcitr06 g09110.1.1 | Dcitr09 g08730.1.1 | Dcitr02 g01530.1.1 | Dcitr09 g02030.1.1 | Dcitr07 g06920.1.1 | Dcitr11 g08810.1.1 | Dcitr00 g06320.1.1 | Dcitr01 g01230.1.1 | Dcitr09 g09620.1.1 |
| Egg *Cmac* CLas– whole body | 27.02 | 128.27 | 134.58 | 1085.07 | 312.06 | 343.23 | 15.59 | 49.46 | 567.45 | 199.16 | 698.5 | 161.78 | 248.69 | 148.17 |
| Nymph *Cmed* CLas+ Low inf whole body | 12.02 | 96.32 | 40.57 | 398.66 | 133.64 | 126.32 | 0 | 43.02 | 1.12 | 124.84 | 416.8 | 119.07 | 160.75 | 0 |
| Nymph *Csin* CLas+ High inf whole body | 8.82 | 89.21 | 39.09 | 334.65 | 131.69 | 102.13 | 0 | 36.49 | 1.4 | 117.74 | 408.72 | 99.61 | 144.58 | 0 |
| Nymph *Csin* CLas– whole body | 12.99 | 99.94 | 49.5 | 312.43 | 161.29 | 134.19 | 0 | 51.94 | 2.57 | 176.32 | 539.91 | 104.1 | 155.61 | 0 |
| Nymph *Cmac* CLas– whole body | 21.77 | 98.95 | 82.02 | 340.27 | 83.11 | 189.27 | 11.78 | 37.44 | 164.67 | 97.37 | 293.69 | 91 | 191.48 | 69.04 |
| Nymph Citrus CLas– whole body | 20.57 | 58.34 | 42.83 | 399.58 | 83.86 | 168.7 | 28.72 | 58.81 | 80.9 | 223.67 | 439.82 | 39.64 | 73.2 | 81.35 |
| Nymph Citrus CLas+ whole body | 4.94 | 38.29 | 59.33 | 621.98 | 146.78 | 160.92 | 2.44 | 53.38 | 190.77 | 444.42 | 766.12 | 44.97 | 115.28 | 114.6 |
| Adult *Cmed* CLas– gut | 38.18 | 289.58 | 200.59 | 647.9 | 426.18 | 605.11 | 26.07 | 50.69 | 381.68 | 296.91 | 1044.09 | 341.38 | 473.57 | 332.39 |
| Adult *Cmed* CLas+ gut | 30.2 | 319.44 | 230.85 | 1705.25 | 428.81 | 659.96 | 27.78 | 60.44 | 529.68 | 352.93 | 1163.03 | 317.72 | 485.17 | 329.78 |
| Adult *Cmed* CLas+ High inf whole body | 30.47 | 131.75 | 59.8 | 406.4 | 151.3 | 171.64 | 0 | 55.4 | 3.76 | 118.88 | 477.59 | 125.92 | 210.48 | 0 |
| Adult *Cmed* CLas+ Low inf whole body | 20.42 | 129.81 | 62.36 | 433.76 | 165.49 | 170.73 | 0 | 43.57 | 3.03 | 137.14 | 519.39 | 105.76 | 181.58 | 0 |
| Adult *Cmed* CLas– whole body | 30.31 | 113.42 | 49.08 | 412.59 | 132.76 | 149.36 | 0 | 71.67 | 2.27 | 70.33 | 411.72 | 118.34 | 191.46 | 0 |
| Adult *Cmac* CLas– whole body | 26.08 | 162.08 | 102.83 | 676.39 | 273.28 | 230.43 | 31.71 | 38.6 | 306.37 | 142.53 | 456.71 | 128.13 | 200.35 | 122.02 |
| Adult Citrus CLas– whole body | 36.98 | 101.48 | 106.15 | 554.39 | 212.37 | 217.42 | 27.13 | 54.78 | 123.29 | 228.76 | 592.56 | 83.13 | 188.23 | 96.36 |
| Adult Citrus CLas+ whole body | 19.8 | 77.29 | 133.73 | 951.84 | 248.6 | 268.61 | 12.34 | 73.2 | 257.5 | 196.34 | 698.74 | 131.62 | 218.34 | 158.58 |
| Adult Citrus CLas– midgut | 103.17 | 399.83 | 371.99 | 2604.59 | 688.74 | 825.5 | 13.38 | 56.53 | 681.66 | 519.97 | 1032.33 | 309.77 | 460.52 | 594.4 |
| Adult Citrus CLas+ midgut | 55.24 | 188.63 | 261.09 | 1449.95 | 529.49 | 549.52 | 8.89 | 33.98 | 732.48 | 484.27 | 1170.81 | 195.92 | 304.17 | 521.08 |
| Adult *Cret* CLas– Female abdomen | 48.78 | 117.05 | 150.69 | 1014.5 | 302.79 | 479.37 | 2.55 | 23.5 | 322.17 | 327.16 | 646.63 | 220.46 | 324.63 | 224.43 |
| Adult *Cret* CLas– Female antennae | 9.84 | 59.74 | 202.33 | 1200.58 | 515.42 | 374.86 | 0.79 | 16.81 | 407.15 | 343.42 | 564.73 | 165.93 | 204.21 | 300.71 |
| Adult *Cret* CLas– Female head | 19.05 | 57.69 | 174.15 | 1299.64 | 416.69 | 357.16 | 1.65 | 23.05 | 452.84 | 326.87 | 606.78 | 173.26 | 258.3 | 321.53 |
| Adult *Cret* CLas– Female leg | 13.96 | 40.8 | 110.91 | 1106.21 | 357.39 | 299.85 | 0.39 | 7.36 | 339.68 | 315.91 | 501.89 | 214.39 | 217.88 | 269.99 |
| Adult *Cret* CLas– Female terminal abdomen | 10.27 | 28.28 | 80.92 | 857.87 | 355.37 | 274.98 | 0.46 | 15.78 | 334.95 | 305.78 | 513.38 | 72.89 | 219.72 | 273.18 |
| Adult *Cret* CLas– Female thorax | 45.76 | 77.36 | 118.42 | 971.53 | 318.3 | 310.44 | 1.83 | 18.82 | 327.81 | 228.02 | 352.69 | 154.87 | 226.17 | 201.78 |
| Adult *Cret* CLas– Male abdomen | 55.04 | 113.66 | 163.22 | 1121.03 | 329.87 | 476.84 | 2.13 | 25.6 | 308.86 | 333.54 | 580.91 | 251.62 | 353.14 | 203.71 |
| Adult *Cret* CLas– Male antennae | 21.34 | 88.88 | 202.61 | 1216.63 | 502.11 | 454.87 | 2.06 | 20.5 | 366.31 | 317.82 | 489.63 | 163.36 | 206.39 | 300.97 |
| Adult *Cret* CLas– Male head | 22.42 | 57.66 | 180.09 | 1215.49 | 397.1 | 382.37 | 1.84 | 22.9 | 427.39 | 340.13 | 591.69 | 156.61 | 223.47 | 333.35 |
| Adult *Cret* CLas– Male leg | 26.24 | 70.25 | 126.32 | 1144.43 | 386.63 | 368.02 | 2.26 | 24.38 | 312.31 | 317.42 | 503.46 | 216.72 | 245.72 | 225.41 |
| Adult *Cret* CLas– Male terminal abdomen | 23.66 | 34.11 | 94.32 | 636.58 | 176.52 | 226.85 | 0.57 | 19.13 | 234.07 | 190.22 | 375.51 | 109.22 | 137.98 | 154.37 |
| Adult *Cret* CLas– Male thorax | 38.48 | 81.1 | 106.97 | 826.51 | 307.15 | 309.28 | 1.82 | 17.83 | 289.69 | 205.61 | 340.62 | 133.14 | 229.48 | 217.37 |
| Adult *Cret* CLas– Female antennae Wu *et al.* [26] | 55.37 | 150.12 | 175.28 | 828.29 | 314.86 | 436.78 | 33.85 | 23.48 | 389.56 | 253.26 | 576.96 | 141.07 | 182.16 | 314.72 |

**Table 5.** (Continued)

| Gene name | V-ATPase a1 | V-ATPase a2 | V-ATPase b | V-ATPase c | V-ATPase e | V-ATPase A | V-ATPase B | V-ATPase C | V-ATPase D | V-ATPase F | V-ATPase G | V-ATPase H (5′) | V-ATPase H (3′) | V-ATPase Ac45 |
|---|---|---|---|---|---|---|---|---|---|---|---|---|---|---|
| Gene ID | Dcitr07 g04330.1.1 | Dcitr07 g01670.1.1 | Dcitr12 g08560.1.1 | Dcitr06 g11110.1.1 | Dcitr03 g19730.1.1 | Dcitr06 g09110.1.1 | Dcitr09 g08730.1.1 | Dcitr02 g01530.1.1 | Dcitr09 g02030.1.1 | Dcitr07 g06920.1.1 | Dcitr11 g08810.1.1 | Dcitr00 g06320.1.1 | Dcitr01 g01230.1.1 | Dcitr09 g09620.1.1 |
| Adult *Cret CLas*− Female terminal abdomen Wu *et al.* [26] | 46.1 | 121.03 | 149.2 | 727.77 | 282.02 | 427.76 | 9.27 | 18.35 | 373.06 | 280.05 | 731.99 | 106.17 | 164.41 | 273.56 |
| Adult *Cret CLas*− Male antennae Wu *et al.* [26] | 48.09 | 190.97 | 202.67 | 923.05 | 349.45 | 524.4 | 8.74 | 35.65 | 448.15 | 268.8 | 602.45 | 147.99 | 218.22 | 369.43 |
| Adult *Cret CLas*− Male terminal abdomen Wu *et al.* [26] | 77.21 | 129.36 | 162.97 | 503.06 | 208.23 | 381.74 | 8.29 | 36.86 | 235.06 | 163.07 | 405.46 | 95.28 | 131.34 | 183.8 |

Comparative expression levels in transcripts per million (TPM) of the *Diaphorina citri V-ATPase* genes encoding the V-ATPase $V_0$ transmembrane, $V_1$ catalytic, and accessory subunits in *D. citri* insects reared on various infected and uninfected citrus varieties. *V-ATPase H* was annotated as two partial models and are both represented here separately as *V-ATPase H* (5′), denoting the 5-prime end of the gene, and *V-ATPase H* (3′), denoting the 3-prime end of the gene. Expression data were collected from the Citrus Greening Expression Network [9], with psyllid RNA-Seq data obtained from NCBI BioProjects PRJNA609978 and PRJNA448935, in addition to several published datasets [25–29]. Citrus hosts are abbreviated as *Csin* (*Citrus sinensis*), *Cmed* (*Citrus medica*), *Cret* (*Citrus reticulata*), and *Cmac* (*Citrus macrophylla*).

**Table 6.** Gene copy comparison of $V_0$ transmembrane subunit *V-ATPase* genes in *Diaphorina citri* and orthologous insect genes.

| Insect | V-ATPase a | V-ATPase b | V-ATPase c | V-ATPase d | V-ATPase e |
|---|---|---|---|---|---|
| *Diaphorina citri* (Hemiptera) | 2 | 1 | 1 | 1 | 1 |
| *Acyrthosiphon pisum* (Hemiptera) | 2 | 1 | 1 | 1 | 1 |
| *Bemisia tabaci* (Hemiptera) | 2 | 1 | 1 | 1 | 1 |
| *Tribolium castaneum* (Coleoptera) | 2 | 2 | 1 | 1 | 2 |
| *Drosophila melanogaster* (Diptera) | 5 | 2 | 1 | 1 | 4 |
| *Aedes aegypti* (Diptera) | 3 | 1 | 1 | 1 | 2 |
| *Apis mellifera* (Hymenoptera) | 2 | 1 | 1 | 1 | 2 |

**Table 7.** Gene copy comparison of $V_1$ catalytic subunit *V-ATPase* genes in *Diaphorina citri* and orthologous insect genes.

| Insect | V-ATPase A | V-ATPase B | V-ATPase C | V-ATPase D | V-ATPase E | V-ATPase F | V-ATPase G | V-ATPase H |
|---|---|---|---|---|---|---|---|---|
| *Diaphorina citri* (Hemiptera) | 1 | 1 | 1 | 1 | 1 | 1 | 1 | 1 |
| *Acyrthosiphon pisum* (Hemiptera) | 1 | 1 | 1 | 2 | 1 | 1 | 2 | 1 |
| *Bemisia tabaci* (Hemiptera) | 1 | 1 | 1 | 1 | 1 | 1 | 1 | 1 |
| *Tribolium castaneum* (Coleoptera) | 1 | 1 | 1 | 3 | 1 | 1 | 2 | 1 |
| *Drosophila melanogaster* (Diptera) | 2 | 1 | 1 | 3 | 1 | 1 | 1 | 1 |
| *Aedes aegypti* (Diptera) | 1 | 1 | 2 | 2 | 1 | 1 | 3 | 1 |
| *Apis mellifera* (Hymenoptera) | 1 | 1 | 1 | 2 | 1 | 1 | 1 | 1 |

**Table 8.** Gene copy comparison of the *Ac45* gene in *Diaphorina citri* and orthologous insect genes.

| Insect | Ac45 |
|---|---|
| *Diaphorina citri* (Hemiptera) | 1 |
| *Acyrthosiphon pisum* (Hemiptera) | 1 |
| *Bemisia tabaci* (Hemiptera) | 1 |
| *Tribolium castaneum* (Coleoptera) | 2 |
| *Drosophila melanogaster* (Diptera) | 2 |
| *Aedes aegypti* (Diptera) | 2 |
| *Apis mellifera* (Hymenoptera) | 1 |

insect. However, it remains unknown at this time whether the elevated expression of *V-ATPase c* relative to other subunits in infected psyllids is associated with higher demand of these proteins in the cell. This should, therefore, be studied further in future research.

Of the $V_1$, catalytic domain, subunit genes, *V-ATPase A* and *V-ATPase B* maintain the highest percentages of sequence identity, consistent with the importance of their function in containing ATP binding sites at the V-ATPase subunits A/B protein interface (Table 3) [32]. Apart from *D. melanogaster*, *V-ATPase A* and *B* also maintain single copies of these two genes across different orders of insects, supporting their conserved nature compared with other genes of this enzyme (Table 7). *V-ATPase A* shows much higher expression than *V-ATPase B* across each measured variable, and *V-ATPase G* shows the highest expression in this group overall (Figure 6a). Unlike *V-ATPase c*, no significant differential expression was observed between the guts of insects reared on infected *versus* uninfected citrus trees (Figure 6b). However, *V-ATPase B* does show a reverse correlation, with a decrease in expression from 28.72 to 2.44 TPM in the whole body of *D. citri* nymphs raised on uninfected *versus* infected *Citrus spp.* (Table 5). A similar expression pattern can be seen throughout many of the *V-ATPase* catalytic genes, which may infer an interaction between these genes and pathogen infection. This warrants further investigations (Table 5).

Figures 2, 3, and 4 depict phylogenetic analyses for the $V_0$ transmembrane and $V_1$ catalytic domains, and the Ac45 protein of V-ATPase, respectively. The individual V-ATPase subunits form clades, regardless of insect species. These clades also have the highest bootstrap values. This agrees with previous research that describes the enzyme as ancient and highly conserved. The evolution of V-ATPase has been analyzed for gene duplication and divergence from other ATP synthases, like F- and A-ATPase, which occur across the three domains of life [3]. Figures 2, 3, and 4 concur and suggest that the V-ATPase enzyme utilized in these insects existed in their common ancestor before they diverged into their respective species. The proteolipid subunit c and subunit d have the shortest branch lengths in Figure 2, consistent with the data shown in Tables 2 and 6, which depict this to be of the most conserved subunits. Subunit c, which is required to form the critical c-ring rotor of V-ATPase [2], and subunit d, which may play a role as part of the central rotor of the V-ATPase [32], appear to have diverged the least compared with the other transmembrane domain subunits and other insect species. In contrast, subunit e has diverged the most (Figure 3). This is consistent with the variable gene copy number observed across different orders of insects and the lower percentages of protein sequence identity seen in *D. citri* pairwise alignments (Tables 2, 6). In addition, the function of subunit e is still unknown for the transmembrane domain subunits [5].

Figure 4 shows the evolutionary relatedness of the *D. citri* Ac45 protein. It is a relatively new protein, critically associated with the assembly of a certain cell type V-ATPase, and is still being studied [6]. For this select group of insect species, Ac45 groups and forms a clade with the other hemipteran protein sequences (Figure 4). *Ac45* is a variable gene when comparing *V-ATPase* across the domains of life, a paralog variability that is also seen among different orders of insects (Table 8) [6, 33]. *Ac45* has diverged the most of all the V-ATPase subunits in *D. citri* compared with other insects. This divergence is seen in phylogenetic analysis, denoted with longer branch lengths (Figure 5), and is also supported in the values of the pairwise alignments, in which the protein shares very little sequence identity across the query lengths (Table 4). Perhaps it is experimentally beneficial that the Ac45 protein shows the least conservation with other insect orthologs. It may serve as a species-specific targeted approach to limiting the psyllid from vectoring the causative agent of citrus greening disease, while leaving related species unharmed and their ecology intact. However, Ac45 shows a markedly depressed transcription level compared with other



subunits (Figure 6a, Table 5). This likely reflects the limits in resolution with current whole RNA isolation and sequencing methods; nevertheless, it still indicates the relatively low total expression. The Ac45 protein has not been observed to exist in every cell type depending on the organism and so is not necessarily utilized by every V-ATPase in the psyllid [6]. Thus, the expression data agree with previously published research.

## CONCLUSION

The V-ATPase is a fundamental enzyme that functions exclusively as an ATP-dependent proton pump in almost every eukaryotic cell. V-ATPase allows for the proper functioning of endosomes and the Golgi apparatus, and it generates a proton-motive force in organelles and across plasma membranes, which is utilized as a driving force for secondary transport processes [1]. Identification of these enzymes in the hemipteran, *D. citri*, provides a novel insect lineage for studies of insect evolution and biology, and may also provide potential targets for *D. citri*-specific molecular mechanisms for the management of HLB in citrus production systems [34–36]. *D. citri* shows no deviation in the expected copy numbers of each of the *V-ATPase* genes (Tables 6–8). The data collected from *D. citri* reveal consistency among the genes previously characterized as highly conserved, such as *V-ATPase c*, *d*, *A*, and *B* (Tables 2–4) [3, 32]. While expression data were not available for *V-ATPase d*, *V-ATPase c* shows comparatively high expression levels overall, and differential expression – 647.9 *versus* 1705.25 TPM – in the guts of adult psyllids fed on uninfected *versus* infected *C. medica* leaves (Figure 5, Table 5). Conversely, the *Ac45* gene shows low expression throughout life stages and tissues compared with other *V-ATPase* genes; however, the highly divergent nature of this gene may serve as a species-specific targeted approach to psyllid control (Table 4, Figure 6).

In hemipterans, RNAi efficacy has been successfully demonstrated for psyllids, whitefly, and leafhoppers [34–41]; planthoppers [42, 43]; bedbugs [44]; and others [45–49]. RNAi specifically targeting the *V-ATPases* in hemipteran insects has been reported for the corn planthopper, *Peregrinus maidis* (Ashmead) (Hemiptera: Delphacidae) [12]; the corn leafhopper, *Dalbulus maidis* (Hemiptera: Cicadellidae) [13]; the brown planthopper, *Nilaparvata lugens* (Stål) (Hemiptera: Delphacidae) [42]; and the bedbug, *Cimex lectularius* L. (Hemiptera: Cimicidae) [44], resulting in increased mortality and reduced fecundity. Thus, the highly divergent nature of these gene sequences provides unique targets that may serve as species-specific targeting for RNAi approaches in the management of psyllid vectors and other hemipteran pests [50, 51].

## REUSE POTENTIAL

The manually curated gene models generated through this *D. citri* community annotation project will be available as part of the official gene set version 3. Analysis of these data, including BLAST and expression profiling, can be conducted using the citrusgreening.org website [52] and Citrus Greening Expression Network (CGEN). The improved annotations presented in this study will facilitate experimental design to investigate the potential of *V-ATPases* as gene targets for therapies to control *D. citri*. Research considering differential expression patterns of V-ATPase transcripts in psyllids fed on *C*Las-infected plants should be conducted. Additional studies are also required to confirm the role of the Ac45 protein, as its divergent nature may provide novel and species-specific gene targets, potentially through the use of RNAi, to control psyllid populations and reduce the effects of pathogens such as *C*Las.

## DATA AVAILABILITY

The datasets supporting this article are available in the *GigaScience* GigaDB repository [24].

## EDITOR'S NOTE

This article is one of a series of Data Releases crediting the outputs of a student-focused and community-driven manual annotation project curating gene models and if required, correcting assembly anomalies, for the *Diaphorina citri* genome project [53].

## DECLARATIONS
## LIST OF ABBREVIATIONS

ADP: Adenosine diphosphate; ATP: Adenosine triphosphate; BLASTp: protein BLAST; *C*Las: *Candidatus* Liberibacter asiaticus; *Cmac*: *Citrus macrophylla*; *Cmed*: *Citrus medica*; *Cret*: *Citrus reticulata*; *Csin*: *Citrus sinensis*; HLB: Huanglongbing; MCOT: Maker, Cufflinks, Oasis, Trinity; NCBI: National Center for Biotechnology Information; QC: Query coverage; CGEN: Citrus Greening Expression Network; RNAi: RNA interference; TPM: transcripts per million; V-ATPase: Vacuolar (H+)-ATP synthase; $V_0$: V-ATPase noncatalytic transmembrane domain; $V_1$: V-ATPase catalytic cytoplasmic domain.

## ETHICAL APPROVAL

Not applicable.

## CONSENT FOR PUBLICATION

Not applicable.

## COMPETING INTERESTS

The authors declare that they have no competing interests.

## FUNDING

This work was supported by USDA-NIFA grants 2015-70016-23028, HSI 2020-38422-32252 and 2020-70029-33199. And NIH grant IDeA P20GM103418.

## AUTHORS' CONTRIBUTIONS

WBH, SJB, TD, and LAM conceptualized the study; TD, SS, TDS, and SJB supervised the study; SJB, TD, SS, and LAM contributed to project administration; RG conducted investigation; PH, MF-G, and SS contributed to software development; PH, MF-G, SS, TDS, and JB developed methodology; SJB, TD, WBH, and LAM acquired funding; RG and CM prepared and wrote the original draft; TD, SJB, SS, TDS, WT, WBH and JB reviewed and edited the draft. All authors read and approved the final version of the manscript.

## ACKNOWLEDGMENT

We would like to thank Helen Wiersma-Koch (Indian River State College) for her assistance.

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
